# COMPOSITE ADVERSARIAL TRAINING FOR MULTIPLE ADVERSARIAL PERTURBATIONS AND BEYOND

## ABSTRACT

One intriguing property of deep neural networks (DNNs) is their vulnerability to adversarial perturbations. Despite the plethora of work on defending against individual perturbation models, improving DNN robustness against the combinations of multiple perturbations is still fairly under-studied. In this paper, we propose composite adversarial training (CAT), a novel training method that flexibly integrates and optimizes multiple adversarial losses, leading to significant robustness improvement with respect to individual perturbations as well as their "compositions". Through empirical evaluation on benchmark datasets and models, we show that CAT outperforms existing adversarial training methods by large margins in defending against the compositions of pixel perturbations and spatial transformations, two major classes of adversarial perturbation models, while incurring limited impact on clean inputs.

## 1  INTRODUCTION

Despite their state-of-the-art performance in tasks ranging from computer vision (Szegedy et al., 2016) to natural language processing (Seo et al., 2017), deep neural networks (DNNs) are inherently susceptible to adversarial examples (Szegedy et al., 2014), which are maliciously crafted samples to deceive target DNNs. A flurry of adversarial attacks have been proposed, which craft adversarial examples via either pixel perturbation (Goodfellow et al., 2015b; Moosavi-Dezfooli et al., 2016; Carlini & Wagner, 2017a) or spatial transformation (Engstrom et al., 2017; Xiao et al., 2018; Alaifari et al., 2019). To defend against such attacks, a line of work attempts to improve DNN robustness by developing new training and inference strategies (Kurakin et al., 2017; Guo et al., 2018; Liao et al., 2018; Tramèr et al., 2018). Yet, the existing defenses are often circumvented or penetrated by adaptive attacks (Athalye et al., 2018), while adversarial training (Madry et al., 2018; Shafahi et al., 2019) proves to be one state-of-the-art defense that still stands against adaptive attacks.

While most adversarial training methods are primarily designed for individual attacks which are either fixed (Madry et al., 2018) or selected from a pre-defined pool (Tramèr & Boneh, 2019; Maini et al., 2020), in realistic settings, the adversary is not constrained to individual perturbation models but free to "compose" multiple perturbation models to construct more powerful attacks. Despite their robustness against individual attacks, the DNNs trained using existing methods often fail to defend against such composite attacks (details in § 2). Moreover, the existing adversarial training methods focus on pixel perturbation-based attacks (e.g., bounded by $\ell_p$-norm balls), while the research on training DNNs robust against spatial transformation-based attacks is still limited.

To bridge this striking gap, in this paper, we present CAT, a novel adversarial training method able to flexibly integrate and optimize multiple adversarial robustness losses, which leads to DNNs robust with respect to multiple individual perturbation models as well as their "compositions". Specifically, CAT assumes an attack model that composes multiple perturbations and, while bounded by the overall perturbation budget, optimally allocates the budget to each iteration. To solve the computational challenges of this formulation, we extend the recent advances on fast projection to $\ell_{p,1}$ mixed-norm ball (Liu & Ye, 2010; Sra, 2012; Béjar et al., 2019) to our setting and significantly improve the optimization efficiency. We validate the efficacy of CAT on benchmark datasets and models. For instance, on MNIST, CAT outperforms alternative adversarial training methods (Tramèr & Boneh, 2019) by over 44% in terms of adversarial accuracy against attacks that combine pixel perturbation and spatial transformation (details in § 4), with comparable clean accuracy and training efficiency.

Our contributions can be summarized as follows. First, we demonstrate that a new class of adversarial attacks, which "compose" multiple perturbations, render most existing adversarial training methods ineffective; then, we propose CAT, the first adversarial training method designed for multiple perturbation models as well as their compositions; further, we validate the efficacy of CAT by comparing it against alternative methods on benchmark datasets and DNNs; finally, we explore the optimization space of composite perturbations, leading to several promising research directions.

## 2 FUNDAMENTALS

### 2.1 ADVERSARIAL TRAINING

Adversarial training is a class of techniques to train robust DNNs by minimizing the worst-case loss with respect to a given adversarial perturbation model. Formally, let $f_\theta$ be a DNN parameterized by $\theta$, $\ell$ the loss function, and $\mathcal{D}_{\text{train}} = \{x_i, y_i\}_{i=1}^n$ the training set. Then the adversarial training with respect to an $\ell_p$ adversary with perturbation magnitude $\epsilon$ is defined as:

$$\theta^* = \arg\min_\theta \sum_i \max_{\delta \in \mathcal{B}_p(\epsilon)} \ell(f_\theta(x_i + \delta), y_i), \tag{1}$$

where $\mathcal{B}_p(\epsilon) = \{\delta : \|\delta\|_p \leq \epsilon\}$ is the $\ell_p$-norm ball of radius $\epsilon$. Here, the inner maximization problem essentially defines the target adversarial attack. For instance, instantiating it as the $\ell_\infty$ projected gradient descent (PGD) attack leads to the well-known PGD adversarial training.

Despite their effectiveness against considered perturbation models (e.g., $\ell_\infty$ perturbation), the existing adversarial training methods often fail to defend against perturbations that they are not designed for (Tramèr & Boneh, 2019). Motivated by this, some recent work explores conducting adversarial training with respect to multiple perturbation models simultaneously.

**AVG and MAX –** Tramèr & Boneh (2019) propose two methods, AVG and MAX, to aggregate multiple perturbations. Specifically, AVG formulates the robustness optimization as:

$$\theta^* = \arg\min_\theta \sum_i \sum_{p \in \mathcal{A}} \max_{\delta_p \in \mathcal{B}_p(\epsilon)} \ell(f_\theta(x_i + \delta_p), y_i) \tag{2}$$

where $\mathcal{A} = \{1, 2, \infty\}$. Compared with Eq. 1, Eq. 2 aggregates multiple adversarial perturbations in the inner loop. Similarly, instead of averaging multiple perturbations, MAX selects the perturbation resulting in the largest loss:

$$\theta^* = \arg\min_\theta \sum_i \max_{\{\delta \in \mathcal{B}_p(\epsilon) | p \in \mathcal{A}\}} \ell(f_\theta(x_i + \delta), y_i) \tag{3}$$

If $\mathcal{A}$ contains only one adversarial perturbation, Eq. 3, Eq. 2, and Eq. 1 are all equivalent.

**MSD –** While AVG and MAX achieve varying degrees of robustness to the considered perturbations, it is practically difficult to minimize the worst-case loss with respect to the union of perturbations. To this end, Maini et al. (2020) propose multiple steepest descent (MSD) which improves MAX (and AVG) along two aspects. First, it selects the largest (or average) perturbation at each inner iteration; Second, it applies the steepest descent instead of the projected gradient descent in generating adversarial inputs. Formally, MSD formulates the optimization at the $t$-th iteration as:

$$\delta_p^{(t+1)} = \text{Proj}_{\mathcal{B}_p(\epsilon)} \left( \delta^{(t)} + v_p(\delta^{(t)}) \right) \quad \text{for } p \in \mathcal{A} \tag{4}$$

$$\delta^{(t+1)} = \arg\max_{\delta_p^{(t+1)}} \ell(f_\theta(x + \delta_p^{(t+1)}), y) \tag{5}$$

where $\text{Proj}_C(\cdot)$ is the projection operator onto the convex set $C$, and $v_p(\delta^{(t)})$ is the steepest descent direction for $\ell_p$ perturbation, $v_p(\delta) = \arg\max_{\|v\|_p \leq \lambda} v^T \nabla \ell(f_\theta(x + \delta), y)$, where $\lambda$ is the step size.

### 2.2 COMPOSITE ADVERSARIAL ATTACK

While the existing adversarial training methods seem effective against individual perturbation models which are either fixed or selected from a pre-defined pool (i.e., the union of perturbations), in

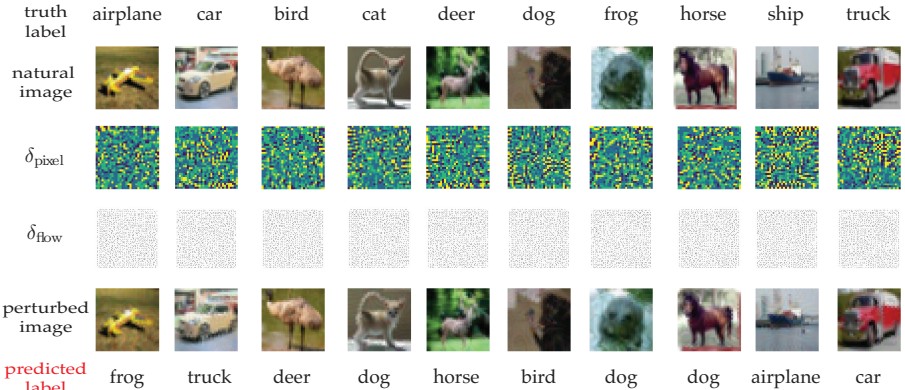

Figure 1: Samples produced by composite attacks on CIFAR10 ($\epsilon_p = 0.015$, $\epsilon_f = 0.175$, untargeted).

a realistic setting, the adversary is able to combine multiple perturbation models to construct more destructive attacks, which we exemplify with a new class of composite adversarial attacks.

Intuitively, the attack constructs an adversarial example by applying a sequence of perturbation models $\{\mathcal{A}_i\}_{i=1}^m$, each $\mathcal{A}_i$ bounded by an independent perturbation budget $\epsilon_i$. Here, we treat $\mathcal{A}_i$ as an abstract operator $\mathcal{A}_i(\cdot, \epsilon_i)$ (e.g., pixel perturbation or spatial transformation), which applies the corresponding perturbation (bounded by $\epsilon_i$) over the output of its previous perturbation:

$$x^{(i)} = x^{(i-1)} + \delta_{i-1}, \quad \delta_i = \mathcal{A}_i(x^{(i)}, \epsilon_i) \quad \text{for } i = 1, \ldots, m \tag{6}$$

We can further generalize the attack to a more flexible setting in which the adversary, while bounded by the overall perturbation budget, is able to optimally allocate the budget to each optimization iteration. The details of this generalization are discussed in Appendix B. Figure 1 illustrates samples generated by the composite adversarial attack which combines one pixel perturbation (with budget $\epsilon_p$) and one spatial transformation (with budget $\epsilon_f$).

Given attacks $\{\mathcal{A}_i\}_{i=1}^m$ with $\mathcal{A}_i$ bounded by $\epsilon_i$, in the composite attack, we re-scale the perturbation budget by a factor of $1/m$ (i.e., $\epsilon_i/m$ for $\mathcal{A}_i$), to make the composition of $\{\mathcal{A}_i\}_{i=1}^m$ comparable with the union attack. Intuitively, with proper setting of $\{\epsilon_i\}_{i=1}^m$, the composition of $\{\mathcal{A}_i\}_{i=1}^m$ is strictly stronger than each individual attack as well as their union (detailed proofs in Appendix A). Figure 2 compares the robust accuracy of AVG, MAX, and MSD on the unions and compositions of multiple perturbations. Observe that while effective on the union attacks, the existing methods fail to defend against the composite attacks, with robust accuracy drop as large as 40%.

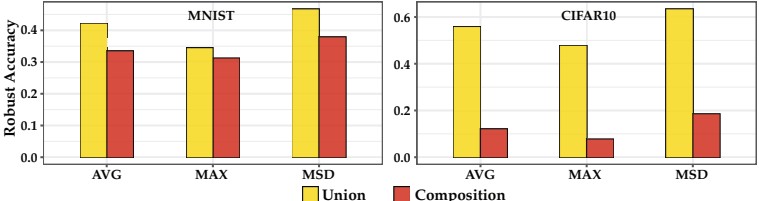

Figure 2: Adversarial accuracy of AVG, MAX, and MSD w.r.t. unions and compositions of $\mathcal{A} = \{\ell_1, \ell_2, \ell_\infty\}$.

## 3 CAT: COMPOSITE ADVERSARIAL TRAINING

We now present CAT, a new adversarial training method to defend against multiple perturbations as well as their compositions.

### 3.1 FORMULATION

At a high level, CAT adopts the composite adversarial attack as the inner loop of Eq. 1, which generates adversarial examples through a sequence of perturbations:

$$x^* = \max_{\{\delta_i\}_{i=1}^m} \ell(f_\theta(x + \sum_{i=1}^m \delta_i), y) \tag{7}$$
$$\text{s.t.} \quad \|\delta_i\| \leq \epsilon_i \quad \text{for} \quad i = 1, \ldots, m$$

As concrete instances, in the case of composing pixel perturbations (i.e., $\ell_1$, $\ell_2$, $\ell_\infty$ perturbations) with budget $\epsilon_1$, $\epsilon_2$, and $\epsilon_\infty$, respectively, the straightforward composite adversarial examples is produced by $x^* = x + \delta_1 + \delta_2 + \delta_\infty$, where we omit the clipping operator.

Next we consider composing pixel perturbation and spatial transformation (Xiao et al., 2018; Alaifari et al., 2019) with budget $\epsilon_p$ and $\epsilon_f$ respectively. We first give a brief introduction of spatial transformation. Instead of directly perturbing the values of pixels, spatial transformation displaces their coordinates. Formally, the input $x$ is represented as a set of tuples $x = \{(u_i, v_i, b_i)\}_{i=1}^n$, where $(u_i, v_i)$ are the coordinates of $x$'s $i$-th pixel and $b_i$ is its value. We set $f = \{(\Delta u_i, \Delta v_i)\}_{i=1}^n$ (with abuse of notations here), where $(\Delta u_i, \Delta v_i)$ are the displacement of $x$'s $i$-th pixel. When we apply $f$ to $x$ to generate the adversarial input $x'$, $(u_i, v_i) = (u_i' + \Delta u_i, v_i' + \Delta v_i)$. Typically, bi-linear interpolation is applied to handle fractional pixel positions (Jaderberg et al., 2015): $b_i' = \sum_{q \in \mathcal{N}(u_i, v_i)} b_q(1 - |u_i - u_q|)(1 - |v_i - v_q|)$, where $\mathcal{N}(u_i, v_i)$ is the neighboring points of $(u_i, v_i)$. Following Alaifari et al. (2019), we measure the perturbation magnitude as $f$'s $\ell_\infty$-norm: $\|f\|_\infty = \max\{\max_i |\Delta u_i|, \max_i |\Delta v_i|\}$.

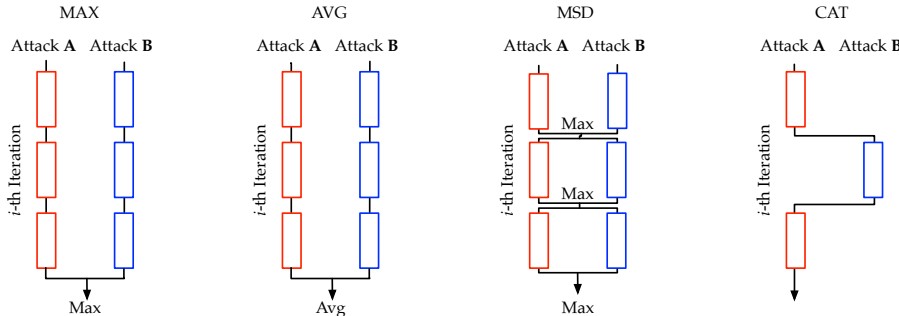

Figure 3: Comparison of different adversarial training frameworks.

## 3.2 DISCUSSION

Figure 3 compares different adversarial training methods (AVG, MAX, MSD, and CAT). The design of CAT enjoys two major benefits. First, by definition, the composite adversarial attack naturally covers the strongest individual attack and the union of these attacks, as demonstrated in the two instantiations above. Second, CAT generalizes adversarial robustness from individual attacks which are either fixed or selected from a fixed pool to their compositions.

As composite adversarial attacks are by nature stronger than individual attacks, setting the perturbation budget overly large in CAT may cause accuracy degradation with respect to clean inputs. We propose a variant $\alpha$-CAT to mitigate this issue. With $0 < 1 \leq \alpha$ as a hyper-parameter, we re-scale the perturbation budget of each component attack $\mathcal{A}_i$ to $\alpha\epsilon_i$ during the adversarial training.

### CAT-r

One issue with the above $\alpha$-CAT formulation is the trade-off between attack strength of individual component attack and clean accuracy. With smaller $\alpha$, we expect a high clean accuracy. However, the component attack might not strong enough to cover the original perturbation size. For instance, if we take $\alpha = \frac{1}{3}$, the trained model may have low robust accuracy for each individual attack and hence their union. On the contrary, a very large $\alpha$ (close to 1) causes significant drop in the clean accuracy, make the robust model is not useful. Our work-round solution for this is to use smaller $\alpha$ to ensure clean accuracy, while we sample component attacks with replacement during adversarial training to enhance the robustness against individual attacks. In fact, under this implementation, we may sample a multiple stack of the same type attack, which corresponds to that attacker with larger perturbation size.

## 4 EMPIRICAL EVALUATION

We empirically evaluate the efficacy of CAT in various settings. All the experiments are performed on MNIST and CIFAR10 dataset. Specifically, our results convey two key messages. First, CAT trains DNNs robust against both composition perturbations and union perturbations in the $\ell_p$ space ($\ell_1$, $\ell_2$ and $\ell_\infty$), suggesting that composite adversarial robustness is a generalization of the adversar-

ial robustness with respect to the union of multiple perturbations. Second, CAT is able to train DNNs robust against both pixel perturbation (e.g., $\ell_\infty$ perturbation) and free-form spatial transformation.

**Models and Hyper-parameters –** On MNIST, we use the network architecture used by both Maini et al. (2020) and Madry et al. (2018), which is a DNN consisting of 4 convolutional layers, followed by 2 fully connected layers. On CIFAR10, we use a pre-activation version of ResNet32 (He et al., 2016), which is build up with 16 residual blocks, followed by 1 global averaging pool layer, and 1 fully connected layer.

**Setting of $\alpha$ –** As discussed in § 3.2, to minimize the performance degradation on clean inputs, we re-scale the allowed perturbation magnitude uniformly by $\alpha$. To set $\alpha$ properly, we conduct a grid search within $\{1.0, 0.9, 0.8, 0.7, 0.6, 0.5\}$ and select the optimal $\alpha$ value by optimizing the trained model's robust accuracy (under the union threat model) and clean accuracy (with less than 10.0% drop from all baselines under the union threat model).

We implement all the algorithms with PyTorch and run all the experiments on a single Nvidia RTX 6000. The detailed setting of models and (hyper-)parameters is summarized in Appendix B.

## 4.1 ROBUSTNESS IN $\ell_p$ SPACE

In the first part, we evaluate CAT and baselines on the regular pixel perturbation-based attacks.

**Baselines –** We compare CAT with MAX- worst-case perturbation, AVG- average-case perturbation (Tramèr & Boneh, 2019), and MSD (Maini et al., 2020). Besides, we also include robust DNNs trained with PGD attacks with respect to individual perturbations.

**Component Attacks for CAT–** We consider 3 commonly used $\ell_p$ perturbations, namely $\ell_\infty$, $\ell_2$, and $\ell_1$ in CAT. For $\ell_\infty$ attack, we use $\ell_\infty$ PGD attack (Madry et al., 2018). For $\ell_2$ attack, we implement an $\ell_2$ PGD adversary. For $\ell_1$ attack, we use the enhanced $\ell_1$ attack proposed in Maini et al. (2020).

**Attacks Used for Evaluation –** To evaluate the robustness of DNNs trained by CAT and baselines, we consider a collection of representative adversarial attacks.

*Individual perturbations –* For $\ell_\infty$ attack, we consider both $\ell_\infty$ PGD attack and Fast Gradient Sign attack (Goodfellow et al., 2015a); for $\ell_2$ attack, we consider $\ell_2$ PGD, DeepFool (Moosavi-Dezfooli et al., 2016), C&W attack (Carlini & Wagner, 2017b), and Salt&Pepper attack (Rauber et al., 2017); for $\ell_1$ attack, we use $\ell_1$ PGD attack.

*Combined perturbations –* Besides, we consider both unions and compositions of multiple perturbations. Following Maini et al. (2020), in the union threat model, the adversary applies all the attacks on the given input and is considered successful if one of the attacks succeeds. In the composite threat model, we consider a set of composite attacks with $\mathcal{A}_1 = \ell_\infty$-PGD, $\mathcal{A}_2 = \ell_2$-PGD, and $\mathcal{A}_3 = \ell_1$-PGD. For the composite attacks, we measure the robust accuracy under different settings of re-scale factor $\alpha$, which is defined similarly in CAT-$\alpha$.

**Results –** Table 1 and Table 2 summarize the results with respect to pixel perturbation-based attacks on CIFAR10 and MNIST. The perturbation budget for each type of $\ell_p$ attack is shown in the tables. To measure models' performance on clean inputs, we use their accuracy on the test set. To evaluate the robustness of each model, we measure the robust accuracy of all the models on a random sample of 1,000 test inputs. The robust accuracy is defined as the fraction of test inputs that are misclassified initially or are predicted to wrong classes after an attack. In reporting robust accuracy, we aggregate results of all the attacks from each target norm. In other word, an input is correctly predicted after attacks for $\ell_p$ norm if and only if all the attacks of $\ell_p$ norms from above fail for the input. Similarly, for the union setting, all the attacks except combined perturbations are considered.

On CIFAR10, with a slight degradation of clean accuracy, CAT achieves the same robustness accuracy as MSD under the union of three $\ell_p$ perturbations and outperforms all other baselines. Furthermore, CAT is much more robust against composite adversarial attacks when $\alpha = 0.5$, outperforming MSD and other baselines by over 10%.

On MNIST, PGD-$\ell_\infty$ ($P_\infty$) achieves the best performance across all the settings, which however is attributed to the holding gradient masking effects (Tramèr & Boneh, 2019). In Appendix C, we show the robust accuracy of models in Table 9 on two decision-based attacks, and we confirm the higher attack success rate with black-box models of $P_\infty$. Thus, we exclude $P_\infty$ in the following

| | $P_\infty$ | $P_2$ | $P_1$ | MAX | AVG | MSD | CAT | CAT-r |
|---|---|---|---|---|---|---|---|---|
| clean accuracy | 83.3% | **90.2%** | 73.3% | 81.0% | 84.6% | 81.1% | 72.6% | 81.6% |
| $\ell_\infty$ attacks ($\epsilon = 0.03$) | **49.7%** | 21.4% | 0.0% | 45.9% | 42.8% | 47.5% | 46.9% | 43.6% |
| $\ell_2$ attacks ($\epsilon = 0.5$) | 59.0% | 65.5% | 0.0% | 62.7% | **67.6%** | 65.8% | 60.3% | 66.7% |
| $\ell_1$ attacks ($\epsilon = 12$) | 16.6% | 25.8% | 10.2% | 39.3% | 57.3% | 54.8% | 59.6% | **63.1%** |
| Union | 16.6% | 18.1% | 0.0% | 34.5% | 42.1% | **46.7%** | **46.7%** | 43.5% |
| Composite (0.5) | 11.2% | 11.6% | 0.3% | 31.0% | 33.5% | 37.7% | **47.0%** | 43.4% |

Table 1. Performances of CAT and baselines on CIFAR10 with $\ell_p$ perturbations ($p = 1, 2, \infty$). Rows represent attacks, and columns denote robust trained models. $P_\infty$, $P_2$, and $P_1$ are models adversarially trained with PGD attacks (with corresponding norms).

| | $P_\infty$ | $P_2$ | $P_1$ | MAX | AVG | MSD | CAT | CAT-r |
|---|---|---|---|---|---|---|---|---|
| clean accuracy | 99.1% | 99.2% | **99.3%** | 98.6% | 99.1% | 98.3% | 91.7% | 98.7% |
| $\ell_\infty$ attacks ($\epsilon = 0.3$) | **92.1%** | 1.5% | 0.0% | 57.5% | 71.0% | 68.1% | 42.5% | 67.1% |
| $\ell_2$ attacks ($\epsilon = 2$) | 63.0% | 75.9% | 45.8% | 68.1% | 71.0% | 74.7% | 67.5% | **77.6%** |
| $\ell_1$ attacks ($\epsilon = 10$) | 71.5% | 69.3% | **77.2%** | 57.1% | 65.1% | 70.9% | 74.3% | 75.2% |
| Union | 59.6% | 1.5% | 0.0% | 47.7% | 55.8% | 63.4% | 42.4% | **63.9%** |
| Composite (0.5) | 49.0% | 3.3% | 0.0% | 7.8% | 12.0% | 18.5% | 38.8% | **49.4%** |

Table 2. Performances of CAT and baseline methods on MNIST with $\ell_p$ perturbations ($p = 1, 2, \infty$). Rows represent attacks, and columns denote robust trained models. $P_\infty$, $P_2$, and $P_1$ are models adversarially trained with PGD attacks (with corresponding norms).

discussion. Across all the other methods, CAT only performs slightly worse than MAX, AVG, and MSD under the union threat model. Under the composite threat model ($\alpha = 0.5$), CAT outperforms baselines in terms of robust accuracy by margins over 20%.

The above results indicate that CAT assumes a strong adversarial attack model during the adversarial training process and produces DNNs with robustness not only against individual perturbations (and their unions) but also against their compositions.

## 4.2 ROBUSTNESS AGAINST PIXEL PERTURBATIONS AND SPATIAL TRANSFORMATIONS

Next we evaluate CAT and baselines against compositions of pixel perturbations and spatial transformations.

**Component Attacks –** For pixel perturbation, we consider $\ell_\infty$ PGD attack. For spatial transformation, we consider a projected gradient descent approach. The key differences between this formulation and ADef (Alaifari et al., 2019) are that 1) PGD is much faster than the DeepFool procedure proposed in ADef and 2) it uses free-form flows instead of the smoothed flows.

**Attacks Used for Evaluation –** We evaluate CAT and baselines against the above individual perturbations as well as their unions and compositions.

**Baselines –** Since MSD is only applicable to pixel perturbation-based attacks, we consider AVG and MAX as the baselines. Plus, we use two DNNs that are adversarially trained using the above piexl perturbation and spatial transformation respectively, which we refer to as $P_\text{pixel}$ and $P_\text{flow}$.

**Results –** Table 3 and 4 summarize the results. We have the following observations. First, CAT achieves similar (on MNIST) or even better (on CIFAR10) robust accuracy than the baselines under the union threat model. Second, CAT outperforms all the baselines under the composite threat model by large margins. For instance, on MNIST, the robust accuracy of all the baselines drops to close to 0 even with $\alpha = 0.5$ (i.e., half of the specified perturbation budget); in contrast, CAT attains 46% robust accuracy.

| | $P_\text{pixel}$ | $P_\text{flow}$ | MAX | AVG | CAT | CAT-r |
|---|---|---|---|---|---|---|
| clean accuracy | **99.1%** | 98.8% | 98.0% | 98.2% | 95.6% | 96.7% |
| pixel attack ($\epsilon = 0.3$) | **92.1%** | 0.0% | 77.3% | 85.2% | 91.8% | 88.9% |
| flow attack ($\epsilon = 0.75$) | 3.0% | 52.4% | 44.8% | 43.3% | 41.3% | **53.5%** |
| Union | 3.0% | 0.0% | 42.5% | 42.4% | 41.2% | **53.5%** |
| Composite (0.5) | 12.5% | 0.0% | 1.0% | 2.0% | 46.0% | **72.4%** |

Table 3. Performance of CAT and baselines on MNIST with respect to pixel and spatial perturbations. Rows represent attacks, and columns denote robust trained models.

| | $P_{\text{pixel}}$ | $P_{\text{flow}}$ | MAX | AVG | CAT | CAT-r |
|---|---|---|---|---|---|---|
| clean accuracy | **83.3%** | 82.5% | 77.2% | 79.8% | 71.3% | 74.7% |
| pixel attack ($\epsilon = 0.03$) | **50.0%** | 0.0% | 45.5% | 45.3% | 47.4% | 39.8% |
| flow attack ($\epsilon = 0.35$) | 21.3% | **48.4%** | 40.8% | 40.6% | 44.1% | 39.0% |
| Union | 18.7% | 0.0% | 35.4% | 32.8% | **38.6%** | 32.3% |
| Composite (0.5) | 26.5% | 0.2% | 38.2% | 36.7% | **43.0%** | 38.6% |

Table 4. Performance of CAT and baselines on CIFAR10 with respect to pixel and spatial perturbations. Rows represent attacks, and columns denote robust trained models.

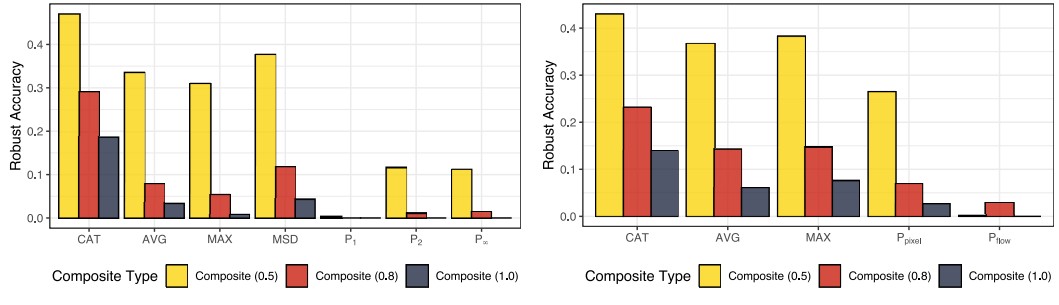

(a) CIFAR10: $\ell_p$ space          (b) CIFAR10: pixel and spatial perturbations

Figure 4: Robust accuracy of CAT and baselines on CIFAR10 dataset under composite adversarial attacks with re-scaling factor $\alpha$=0.5, 0.8, and 1.0.

### 4.3 ROBUSTNESS AGAINST COMPOSITE ATTACKS WITH VARYING $\alpha$

Thus far, we have assumed the defender and attacker use the same setting of re-scaling factor ($\alpha = 0.5$) in attacking the DNNs. Next, we evaluate the impact of varying $\alpha$ by the attacker on the robust accuracy. Figure 4 summarizes the results on CIFAR10 under $\alpha = 0.5$, 0.8, and 1.0 (which correspond to stronger attacks). Observe that as expected, both CAT and baselines experience performance degradation under large $\alpha$ and yet, CAT still consistently outperforms all the baselines by large margins across all the settings.

## 5 EXPLORING THE SPACE OF COMPOSITE PERTURBATIONS

In this section, we empirically study the critical properties of composite adversarial attacks and CAT. One particular aspect is the impact of the ordering of component attacks on the adversarial training. Besides, we consider an even stronger composite attack which, bounded by overall perturbation budget, is able to optimally allocate the budget to each round.

### 5.1 ORDERING OF COMPONENT ATTACKS

We evaluate the impact of the ordering of component attacks under the setting of $\ell_p$ perturbations only as well as compositions of pixel perturbations and spatial transformations.

$\ell_p$ **Perturbations –** In this set of experiments, we consider the compositions of $\ell_\infty$-PGD and $\ell_2$-PGD attacks. Table 5 and 6 summarize the performance of CAT under the compositions of $\ell_\infty$ and $\ell_2$ perturbations. We have the following observations. First, flipping the two attacks in CAT has little impact on both clean accuracy and robust accuracy of CAT. Second, observable from the last two columns of each table, the effectiveness of composite adversarial attacks seems also independent of the ordering of component attacks.

| | Clean Accuracy | $\ell_\infty$ | $\ell_2$ | Union | $\ell_\infty$ - $\ell_2$ (0.5) | $\ell_2$ - $\ell_\infty$ (0.5) |
|---|---|---|---|---|---|---|
| $\ell_\infty = 0.3$ - $\ell_2 = 2.0$ | 99.1% | 92.5% | 77.7% | 77.4% | 84.2% | 84.2% |
| $\ell_2 = 2.0$ - $\ell_\infty = 0.3$ | 99.1% | 92.8% | 79.5% | 79.1% | 85.7% | 84.7% |
| $\ell_\infty = 0.15$ - $\ell_2 = 2.0$ | 99.4% | 96.9% | 58.2% | 58.2% | 86.0% | 85.7% |
| $\ell_2 = 2.0$ - $\ell_\infty = 0.15$ | 99.4% | 96.8% | 57.7% | 57.7% | 85.3% | 85.3% |
| $\ell_\infty = 0.3$ - $\ell_2 = 1.0$ | 99.1% | 92.4% | 95.9% | 92.1% | 93.3% | 93.4% |
| $\ell_2 = 1.0$ - $\ell_\infty = 0.3$ | 99.0% | 92.2% | 95.8% | 92.0% | 93.4% | 93.8% |

Table 5. Impact of the ordering of component attacks in CAT on MNIST under compositions of $\ell_\infty$ and $\ell_2$ perturbations ($\mathcal{A}_1$ - $\mathcal{A}_2$ is the ordering used in CAT and composite attacks).

| | Clean Accuracy | $\ell_\infty$ | $\ell_2$ | Union | $\ell_\infty$ - $\ell_2$ (0.5) | $\ell_2$ - $\ell_\infty$ (0.5) |
|---|---|---|---|---|---|---|
| $\ell_\infty = 0.03$ - $\ell_2 = 0.5$ | 82.7% | 48.8% | 64.4% | 48.8% | 57.0% | 57.0% |
| $\ell_2 = 0.5$ - $\ell_\infty = 0.03$ | 82.8% | 50.3% | 64.9% | 50.3% | 58.5% | 58.6% |
| $\ell_\infty = 0.015$ - $\ell_2 = 0.5$ | 88.8% | 69.1% | 68.2% | 67.1% | 68.3% | 68.4% |
| $\ell_2 = 0.5$ - $\ell_\infty = 0.015$ | 89.1% | 69.8% | 67.2% | 66.7% | 68.7% | 68.7% |
| $\ell_\infty = 0.03$ - $\ell_2 = 0.25$ | 83.1% | 50.0% | 74.5% | 50.0% | 62.7% | 62.5% |
| $\ell_2 = 0.25$ - $\ell_\infty = 0.03$ | 82.7% | 51.0% | 73.4% | 51.0% | 62.7% | 62.7% |

Table 6. Impact of the ordering of component attacks in CAT on CIFAR10 under compositions of $\ell_\infty$ and $\ell_2$ perturbations ($\mathcal{A}_1$ - $\mathcal{A}_2$ is the ordering used in CAT and composite attacks).

| | | Clean Accuracy | Pixel | Flow | Union | $p_\infty$ - $f_\infty$ (0.5) | $f_\infty$ - $p_\infty$ (0.5) |
|---|---|---|---|---|---|---|---|
| MNIST | $p_\infty$ - $f_\infty$ | 95.6% | 91.8% | 41.3% | 41.2% | 46.0% | 52.6% |
| | $f_\infty$ - $p_\infty$ | 96.7% | 92.8% | 34.4% | 34.4% | 40.6% | 42.7% |
| CIFAR10 | $p_\infty$ - $f_\infty$ | 71.3% | 47.4% | 44.1% | 38.3% | 43.0% | 42.3% |
| | $f_\infty$ - $p_\infty$ | 70.3% | 48.0% | 43.2% | 38.8% | 42.8% | 42.6% |

Table 7. Impact of the ordering of component attacks (pixel and spatial perturbations) in CAT on MNIST and CIFAR10 ($\mathcal{A}_1$ - $\mathcal{A}_2$ denotes the ordering of component attacks used in CAT and composite attacks, while the perturbation budget is the same as Table 3 and 4).

**Pixel and Spatial Perturbations –** Similarly, we empirically evaluate the impact of the ordering of pixel and spatial perturbations. Here we follow the same setup as §4 (the same attacks and $\alpha$ for each dataset). The results are summarized in Table 7. We observe that on CIFAR10, the ordering has fairly limited impact as in the $\ell_p$ case; however, we find that first applying pixel perturbation and then spatial transformation results in more robust model on MNIST. We consider the study of the root cause of this interesting phenomenon as our ongoing work.

## 5.2 ROBUSTNESS AGAINST FINE-GRAINED COMPOSITE ADVERSARIES

Finally, we consider an extension of the basic composite adversarial attack (each perturbation is applied only once) to a multiple round setting. Under this setting, the adversary, while bounded by the overall perturbation budget, is able to optimally allocate the budget to each iteration, leading to even stronger attacks.

**Formulation and Solution –** We sketch the differences between CAT and $K$-round CAT here, with full details deferred to the appendix B.

*Perturbation Accounting –* For each component perturbation $\mathcal{A}_i$, instead of seeking a single $\delta_i$, we divide it into $K$ parts $\delta_{i,k}$ for $k = 1, \ldots, K$. We measure the overall perturbation cost as the sum of $\{\|\delta_{i,k}\|_p\}_{k=1}^K$, where $p$ is $\ell_p$ norm. In other word, the new constraint is $\sum_{k=1}^K \|\delta_{i,k}\|_p \le \epsilon_i$.

*Optimization Solution –* This new constraints essentially specifies a $\ell_{1,p}$ mixed-norm ball constraint. Therefore, we can still solve this new optimization problem with projected gradient descent. We extend the recent advances (Liu & Ye, 2010; Béjar et al., 2019) to solve this new problem for the cases of $p = 2$ and $p = \infty$ respectively (details in the appendix B).

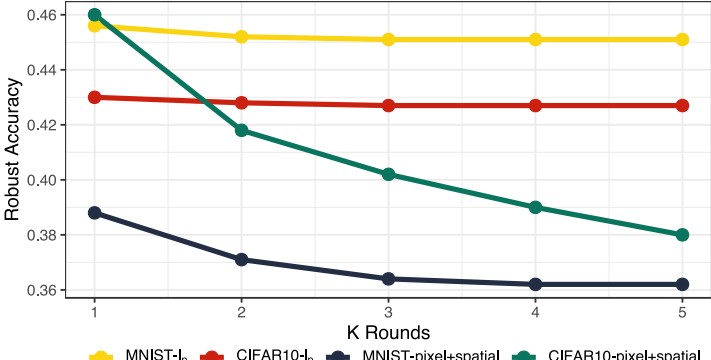

Figure 5: Robust accuracy versus the number of rounds $K$ on MNIST and CIFAR10 under composite adversarial attacks.

**Results –** Figure 5 displays how $K$-round composite attack impacts the robust accuracy of CAT under the settings of $\ell_p$ perturbations only as well as pixel plus spatial perturbations. As $K$ increases, the robust accuracy decreases by 4% and 7% respectively on MNIST and CIFAR10 under the setting

of pixel plus spatial perturbations, while the decrease is much less evident under the setting of $\ell_p$ perturbations only, indicating that CAT is fairly robust to $K$-round composite attacks. Moreover, by incorporating $K$-round composite attacks in the adversarial training of CAT, we expect to see further robustness improvement.

## 6 CONCLUSION

While effective against individual perturbation models, existing adversarial defenses often fail to defend against combinations of multiple perturbations. In this paper, we first present a new class of composite attacks that combine multiple perturbations and penetrate the state-of-the-art defenses. We then propose composite adversarial training (CAT), a novel training method that improves DNNs robustness not only against individual perturbations but also against their compositions. Empirical evaluation on benchmark datasets and models shows its promising performance.

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

## A   PROOFS

We assume a set of component models $\{\mathcal{A}_i\}_{i=1}^m$, each $\mathcal{A}_i$ bounded by budget $\epsilon_i$.

**Lemma 1.** *The union of $\{\mathcal{A}_i\}_{i=1}^m$ is stronger than each individual attack.*

*Proof.* (Lemma 1) The adversarial loss of a given input $(x, y)$ with respect to $\mathcal{A}_i$ is defined as:

$$\mathcal{L}_i(x) = \max_{\delta \in \mathcal{B}_i(\epsilon_i)} \ell(f(x + \delta), y) \tag{8}$$

where $\mathcal{B}_i(\epsilon)$ is the feasible set for $\mathcal{A}_i$ with bound $\epsilon_i$. Meanwhile, the adversarial loss with respect to the union of $\{\mathcal{A}_i\}_{i=1}^m$ is defined as:

$$\mathcal{L}_u(x) = \max_i \max_{\delta \in \mathcal{B}_i(\epsilon_i)} \ell(f(x + \delta), y) \geq \mathcal{L}_i(x) \tag{9}$$

Given that $\mathcal{A}_i$ is arbitrarily chosen, the union attack is stronger than each component attack. □

Therefore, we only need to show that the composite attack is stronger than the union attack. We prove this by showing that the feasible set for the composite attack is larger than the union attack. To simplify the discussion, we consider two $\ell_p$ attacks $\mathcal{A}_1$ and $\mathcal{A}_\infty$ with budget $\epsilon_1$ and $\epsilon_\infty$ respectively, and $d$ is the data dimensionality.

**Lemma 2.** *If $d(\epsilon_1 + \epsilon_\infty)^d / 2^d - 2(\epsilon_1 - \epsilon_\infty)^d > \epsilon_1^d / 2^d + d\epsilon_\infty^d$, the composition of $\mathcal{A}_1$ and $\mathcal{A}_\infty$ is strictly stronger than their union.*

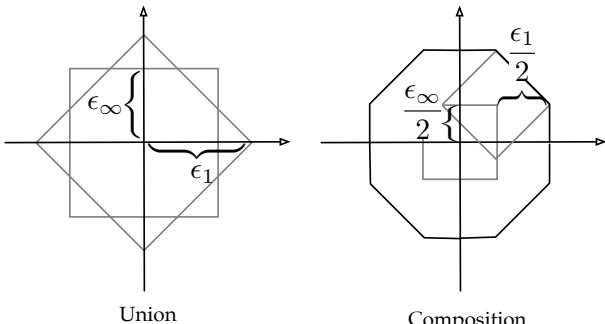

Union                    Composition

Figure 6: Comparison of union and composition for $d = 2$.

*Proof.* (Lemma 2) First, it can be verified that with the constraint, $\epsilon_\infty < \epsilon_1 < d\epsilon_\infty$, thus the $d$-dimensional hypercubes represented by $\epsilon_1$ and $\epsilon_\infty$ intersect but none of them completely contains the other. Figure 6 shows the case for $d = 2$.

The feasible set $\mathcal{B}_u$ for the union attack is given by:

$$\mathcal{B}_u = \mathcal{B}_1(\epsilon_1) \cup \mathcal{B}_\infty(\epsilon_\infty) \tag{10}$$

where $\mathcal{B}_p(\epsilon_p)$ denotes the feasible set for $\mathcal{A}_p$ $(p = 1, \infty)$. The volume of $\mathcal{B}_u$, $\mathrm{Vol}(\mathcal{B}_u)$ is given by:

$$\mathrm{Vol}(\mathcal{B}_u) = 2^d \epsilon_\infty^d + \frac{2^{d+1}}{d}(\epsilon_1 - \epsilon_\infty)^d \tag{11}$$

In comparison, the volume of the feasible set $\mathcal{B}_c$ for the composite attack (with budget $\epsilon_1/2$ and $\epsilon_\infty/2$ for $\mathcal{A}_1$ and $\mathcal{A}_\infty$) is given by:

$$\mathrm{Vol}(\mathcal{B}_c) = (\epsilon_1 + \epsilon_\infty)^d - \epsilon_1^d / d \tag{12}$$

Given the constraint, it is trivial to see $\mathrm{Vol}(\mathcal{B}_c) > \mathrm{Vol}(\mathcal{B}_u)$, indicating that the composite attack entails a larger perturbation space than the union attack. □

This result can be generalized to the cases of other $\ell_p$ attacks and more than two component attacks.

## B  EXPERIMENT SETTINGS

We present the detailed setting of CAT, baseline methods, as well as the attacks used in § 4.

| Case | MNIST-$\ell_p$ | CIFAR10-$\ell_p$ | MNIXT-pixel and spatial | CIFAR10-pixel and spatial |
|---|---|---|---|---|
| $\alpha$ | 0.8 | 0.8 | 1.0 | 0.7 |

Table 8. The rescale factor $\alpha$ for CAT models presented in the paper.

## B.1 MODEL TRAINING

We reuse a few pre-trained models from repository provided by previous work. For the rest of models, their training use the following setups for each dataset.

- **MNIST**. The models are optimized with an Adam optimizer (Kingma & Ba, 2015). The learning rate linearly increases from 0 to 0.001 in the first 6 epochs, and then linearly decreases to 0 in the last 9 epochs.

- **CIFAR10**. The models are optimized by a SGD optimizer with momentum of 0.9. The learning rate linearly increases from 0 to 0.1 in the first 20 epochs, then linearly decrease to 0.005 in the next 20 epochs, and it linearly decay to 0 in the last 10 epochs. Besides, we regularize models with a $\ell_2$ weight decay of $5 \times 10^{-4}$.

## B.2 CAT

- $\ell_p$ **cases**. We run the composite attack during CAT model training with step sizes of $0.1 \times \epsilon_p$ for each $p \in \{1, 2, \infty\}$. The number of iterations for attacks is 50.

- **pixel and spatial perturbations**. We run the composite attack during CAT model training with step sizes of $0.1 \times \epsilon_p$ and $0.1 \times \epsilon_f$ for pixel and spatial perturbation. The number of iterations for attacks is 40.

We also present the re-scaling factor $\alpha$ for the models presented in the main text in Table 8.

## B.3 ADVERSARIAL ATTACKS

In evaluation, we run all the attacks except C&W with 5 random restarts. Below we summarize all the attacks we used in the evaluation and their hyper-parameters.

- **Attacks from Foolbox**. We use the following attacks from Foolbox 3.1.1[1]: $\ell_\infty$ PGD, $\ell_2$ PGD, $\ell_1$ PGD, Fast Gradient Sign Method, DeepFool, C&W, Salt&Pepper. We take their default settings from Foolbox in the evaluation.

- **Our Implementation of $\ell_p$ PGD Attacks**. Plus, we also implement PGD attacks for three $\ell_p$ norms ourselves, which achieves higher attack succces rate than the version from the Foolbox. For $\ell_\infty$ PGD, we run 200 iterations with a step size of $0.1 \times \epsilon$. For $\ell_2$ PGD, we run 500 iterations with a step size of $0.05 \times \epsilon$. Our implementation of $\ell_1$ PGD attack is based upon Section A.1 of (Maini et al., 2020), where we set the the range of number of pixels to modify in each iteration $[k_1, k_2]$ as: $k_1 = 5$ and $k_2 = 20$. The number of iterations of this attack is 200, and the step size is set to $0.05 \times \epsilon$.

- **The PGD Spatial Perturbation Attack**. We run this attack with 200 iterations and step size of $0.1 \times \epsilon_f$.

- **Composite Adversarial Attacks**. We run this attack with 200 iterations and step sizes of $0.1 \times \epsilon$ for all the component attacks.

## B.4 BASELINES

- $\ell_p$ **Perturbations**. We use pre-trained robust models provided by MSD (Maini et al., 2020)[2] for all the baseline methods, including $P_\infty$, $P_2$, $P_1$, AVG, MAX, and MSD.

- **Pixel and Spatial Perturbations**. The adversarial trained models with $\ell_\infty$ PGD is the same as in the $\ell_p$ perturbations case. For $P_{\text{flow}}$, the number of iterations for spatial attack is 50, and the step size is set to $0.1 \times \epsilon_f$. The same rule applies to AVG and MAX on the both two attacks.

---

[1]https://github.com/bethgelab/foolbox
[2]https://github.com/locuslab/robust_union

## C  $K$-ROUND COMPOSITE ADVERSARIAL ATTACK

We describe the detailed formulation for the multiple round composite adversarial attack and technical tools to solve this new attack.

### C.1  FORMULATION

We denote $m$ component attacks as $\mathcal{A}_1, \ldots, \mathcal{A}_m$, and whose perturbation sizes are $\epsilon_1, \ldots, \epsilon_m$ respectively. Unlike the definition in §2, here we represent each attack $\mathcal{A}_i$ as $\mathcal{A}_i(x, \delta_i)$, which denotes applying perturbation $\delta_i$ onto $x$ with the mechanism of $\mathcal{A}_i$. The $K$-round attack generalizes composite adversarial attack in the following sense, the attack runs in $K$-round. At $k$-th round, it performs a composite attack with perturbations $\delta_{k,1}, \ldots, \delta_{k,m}$. The constraint for adversary is the overall magnitude of perturbations he spent for each attack. Formally,

$$(\delta_1^*, \ldots, \delta_m^*) \arg\min_{\boldsymbol{\delta}_i} \ell(x_K, y)$$

$$\text{s.t. } \begin{cases} x_0 = x \\ x_k = \mathcal{A}_m \left( \ldots \mathcal{A}_2 \left( \mathcal{A}_1(x_{k-1}, \delta_{k,1}), \delta_{k,2} \right) \ldots, \delta_{k,m} \right) \quad k = 1, \ldots, K \\ \sum_{k=1}^{K} \|\delta_{i,k}\| \leq \epsilon_i \quad i = 1, \ldots, m \end{cases} \quad (13)$$

where $x_k$ is the perturbed sample after the $k$-th round and $\boldsymbol{\delta}_i = (\delta_{i,1}, \ldots, \delta_{i,K})$ is the concatenation of the perturbation at each iteration. Observe that by specifying $K$, Eqn 13 instantiates a spectrum of attacks. As $K$ approaches infinity, Eqn 13 essentially considers all finite combinations of ways of allocating the total budgets $\{\epsilon_i\}_{i=1}^{m}$ using the $m$ perturbation mechanisms.

### C.2  DERIVATION

Now we present an iterative projected gradient descent algorithm to find a solution to Eqn (13). We use the superscript $(t)$ to denote the value of related variables at $t$-th iteration. We randomly initialize $\delta_i^{(0)}$. At the $t$-th iteration, the update rule is defined as:

$$g_i^{(t)} \leftarrow \frac{\partial \ell(x_K^{(t)}, y)}{\partial \delta_i^{(t)}} \quad (14)$$

$$\delta_i^{(t+1)} \leftarrow \Pi_{\{\delta_i : \sum_{k=1}^{K} \|\delta_{i,k}\| \leq \epsilon_i\}} (\delta_i^{(t)} - \alpha g_i^{(t)}) \quad (15)$$

where $\alpha$ is the learning rate and $\Pi_S(\cdot)$ is the projection operator of a convex set $S$. The computation of Eqn (14) is straightforward. We focus our discussion on the projection operator in Eqn (15).

Without loss of generality, we omit the subscript $i$ for simplicity. Let $V$ be a matrix with its $k$-th row as $\delta_k$ (for $k = 1, \ldots, K$). Then the summation $\sum_{k=1}^{K} \|\delta_k\| \leq \epsilon$ can be rewritten as $\sum_{k=1}^{K} \|v_k\|$. Additional, We suppose the perturbation of $i$-th component attack is measured with $\ell_p$ norm for some $p \in \{1, 2, \infty\}$, which hold for all the experiments in the paper. Thus, we reach to the $\ell_{p,1}$ mixed-norm of $V$, denoted as $\|V\|_{p,1}$, which is a special case of $\ell_{p,q}$ mixed-norm ball: $\|V\|_{p,q} = (\sum_{i=1}^{m} \|v_i\|_p^q)^{1/q}$. Hence, we cast Eqn (15) as the projection operator onto an $\ell_{p,1}$ mixed-norm ball. For $p = 1$, the calculation is straightforward since it reduces to the $\ell_1$ projection operator for the concatenated vectors. We work with $p = 2$ and $p = \infty$ in the next.

### C.3  IMPLEMENTATION FOR $p = 2$

We leverage Algorithm 1 in (Sra, 2012) to compute the $\ell_{2,\infty}$ mixed-norm of an input $x$. Interested readers could find more details in (Sra, 2012).

### C.4  IMPLEMENTATION FOR $p = \infty$

We leverage the method proposed in (Béjar et al., 2019), which solves the proximal operator of $\ell_{1,\infty}$ mixed-norm using an active set approach and attains better efficiency than previous methods (Quattoni et al., 2009; Gustavo et al., 2018; Chau et al., 2019). However, as this is a primitive procedure

in CAT, which is executed for hundreds of iterations, the basic implementation in (Béjar et al., 2019) is not scalable for our setting.

We now improve the scalability of (Béjar et al., 2019) for CAT, with their original algorithm sketched in Algorithm 1. The method is based on computing the proximal operator of the dual form of $\ell_{\infty,1}$ mixed-norm ball, $\ell_{\infty,1}$ mixed-norm. We define $U = \text{abs}(V)$. Given an initial radius of $\ell_1$ ball for its row $u_i$, denoted by $r$, it iteratively improves $r$ by first projecting each row with $\ell_1$-norm exceeding $r$ to the $\ell_1$ ball of radius $r$ (line 4 to 8) and then computing a larger $r$ based on the nonzero elements of the projected row (line 9). In particular, Algorithm 1 uses sorting to solve the $\ell_1$-norm projection problem, which is detailed in Algorithm 2. We state our more efficient implementation based on two key observations here.

**Observation 1 –** The projection radius $r$ (line 6) in Algorithm 1 for related rows increases at every iteration. Thus, for each related row $v$, we face a set of queries with increasing radii $r_1 \leq \cdots \leq r_T$, where $T$ is the number of queries. Following the notations of Algorithm 2, let $u$ denote the sorted $v$ in non-decreasing order, and we define $h_k = \sum_{j=1}^{k} u_j - ku_k$. Note that $h$ is monotonically increasing:

$$h_{k+1} - h_k = u_{k+1} - (k+1)u_{k+1} + ku_k = k(u_k - u_{k+1}) \geq 0$$

Leveraging this observation, we optimize Algorithm 2 as follows. At line 2, $K$ is the largest item among $1, \ldots, N$ such that $r > h_K$; we only need to scan $h_k$ one pass to find the optimal $K_t$ for all $a_t$ due to that both $h$ and $r$ are increasing. At line 3, we only need to pre-process the partial sums once for each row before the main loop.

**Observation 2 –** To update the radius $r$ of the $\ell_1$-norm ball in Algorithm 1, we need to access the statistics of line $7 \sim 9$. A critical observation is that it is not necessary to explicitly compute the projected vector at every iteration to compute the statistics. Specifically, line 7 computes the number of non-zero elements $\mathcal{J}_i$ for the projected $i$-th row; and line 9 updates $r$ using the sums of the projected rows $\sum_{j=1}^{n} x_{i,j}$ and $\mathcal{J}_i$. Combining the previous observation, if $K_{i,t}$ is the optimal $K$ (line 2 of Algorithm 2) for the $t$-th radius $r_t$ and the $i$-th row, it holds that

$$|\mathcal{J}_i| = K_{t,i} \text{ and } \sum_{j \in \mathcal{J}_i} u_{i,j} = \sum_{\{j:x_{i,j}>0\}} (x_{i,j} + \tau_{t,i}) = r_t + K_{i,t}\tau_{i,t}$$

where $\tau_{i,t}$ is calculated based on line 3 of Algorithm 2 for $i$-th row with radius $r_t$. Thus, we can avoid computing the intermediate $\ell_1$ projections.

---

**Algorithm 1:** Proximal operator of mixed $\ell_{1,\infty}$ norm: $\text{prox}_{\lambda \|\cdot\|_1}(\cdot)$ Béjar et al. (2019)

---
**Input:** $m \times n$ matrix $V$
**Output:** $X$
1  $U \leftarrow \text{abs}(V)$ ;
2  $r \leftarrow$ initial radius computed via Lemma 2 and Lemma 3 in Béjar et al. (2019) ;
3  **do**
4     $\mathcal{M} \leftarrow \{i | r < \|u_i\|_1\}$ ;
5     **foreach** $i$ in $\mathcal{M}$ **do**
6        $x_i \leftarrow \Pi_{\|\cdot\|_1 \leq r}(u_i)$ ;
7        $\mathcal{J}_i \leftarrow \{j | x_{i,j} > 0\}$ ;
8     $r \leftarrow \frac{\sum_{i \in \mathcal{M}} \frac{1}{|\mathcal{J}_i|} \sum_{j \in \mathcal{J}_i} u_{i,j} - \lambda}{\sum_{i \in \mathcal{M}} \frac{1}{|\mathcal{J}_i|}}$ ;
9  **while** $\mathcal{M}$ or $\{\mathcal{J}_i\}_{i=1}^{m}$ change;
10  **for** $i \leftarrow 1, \ldots, m$ **do** $\mu_i \leftarrow \max\left(\frac{\sum_{j \in \mathcal{J}_i} u_{i,j} - r}{\lambda |\mathcal{J}_i|}, 0\right)$;
11  $X \leftarrow \text{sgn}(V) \odot \max\left(U - \lambda \mu \mathbf{1}^T, \mathbf{0}\right)$ ;

---

## D  ADDITIONAL RESULTS

### D.1  DECISION-BASED ATTACKS ON MNIST

Table 9 summarizes the results of all the models in Table 2 on two decision-based attacks: $\ell_2$-Pointwise attack and $\ell_1$-Pointwise attack (Schott et al., 2019). One may notice that the inferior performance of $P_\infty$ model on these black-box attacks compared to its performance in Table 2 for white-box attacks. In summary, we find strong gradient masking effects within this model.

---

**Algorithm 2:** Algorithm for projection $\boldsymbol{v} \in \mathbb{R}^N$ onto simplex $\sum_{n=1}^N x_n = r$, and $x_n \geq 0$ for $n = 1, \cdots, N$.

---

**Input:** $\boldsymbol{y} \in \mathbb{R}^N$
**Output:** $\boldsymbol{x}$

1   $\boldsymbol{u} \leftarrow$ sort $\boldsymbol{v}$ in non-increasing order: $u_1 \geq \cdots \geq u_N$ ;
2   $K \leftarrow \max_{1 \leq k \leq N} \{k | (\sum_{j=1}^k u_j - r)/k < u_k\}$ ;
3   $\tau \leftarrow (\sum_{k=1}^K u_k - r)/K$ ;
4   **for** $n \leftarrow 1, \cdots, N$ **do**
5     $\left\lfloor \quad x_n \leftarrow \max(v_n - \tau, 0) \right.$ ;

|  | $P_\infty$ | $P_2$ | $P_1$ | MAX | AVG | MSD | CAT |
|---|---|---|---|---|---|---|---|
| Pointwise ($\ell_2$) | 68.9% | 97.3% | 98.4% | 94.0% | 92.9% | 95.3% | 87.5% |
| Pointwise ($\ell_1$) | 22.5% | 92.7% | 93.9% | 83.4% | 73.0% | 86.6% | 82.0% |

Table 9. Robust Accuracy to decision-based attacks for models trained with CAT and baseline methods on MNIST dataset.

### D.2   VISUALIZATION OF COMPOSITE ADVERSARIAL EXAMPLES ON MNIST

Figure 7 complements Figure 1 with adversarial examples against a natural model for MNIST dataset.

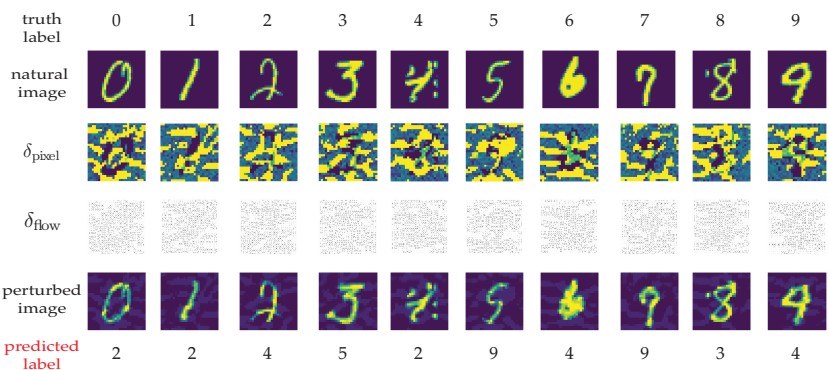

Figure 7: Samples produced by composite attacks on MNIST. ($\epsilon_p = 0.09, \epsilon_f = 0.225$, untargeted).

### D.3   PRELIMINARY RESULTS FOR CIFAR 100 DATASETS

Table 10 and Table 11 present performances of baseline models and CAT for CIFAR100 dataset. Similar to previous results, with slightly reduction in clean accuracy, CAT outperforms MAX and AVG as well as simple PGD adversarial trained models. The MSD is excluded here temporally due to it takes extreme long time to train. We will fill MSD's result when it is available.

### D.4   COMPUTATIONAL EFFICIENCY OF CAT

We describe the computational cost of MAX, AVG, and CAT here. MSD is excluded here due to it requires way more time, as the official implementation. We consider filling its result with our implementation later. All the experiments are with CIFAR10 dataset and of $\ell_p$ settings. The batch size of all the experiments is 50, and the total number of epochs is 50. The result is in Table 12.

|  | $P_\infty$ | $P_2$ | $P_1$ | MAX | AVG | CAT |
|---|---|---|---|---|---|---|
| clean accuracy | 49.0% | **70.7%** | 64.5% | 56.7% | 62.8% | 53.3% |
| $\ell_\infty$ attacks ($\epsilon = 0.03$) | **25.9%** | 2.8% | 0.0% | 23.6% | 17.8% | 22.6% |
| $\ell_2$ attacks ($\epsilon = 0.5$) | 34.5% | 25.6% | 0.0% | 36.8% | **38.7%** | 38.6% |
| $\ell_1$ attacks ($\epsilon = 12$) | 9.6% | 5.0% | 0.0% | 29.6% | 36.7% | **37.6%** |
| Union | 9.2% | 2.4% | 0.0% | 21.8% | 17.7% | **22.3%** |
| Composite (0.5) | 8.3% | 1.6% | 0.0% | 19.3% | 15.8% | **22.7%** |

Table 10. Performances of CAT and baselines on CIFAR100 with $\ell_p$ perturbations ($p = 1, 2, \infty$). Rows represent attacks, and columns denote robust trained models. $P_\infty$, $P_2$, and $P_1$ are models adversarially trained with PGD attacks (with corresponding norms).

|  | $P_{\text{pixel}}$ | $P_{\text{flow}}$ | MAX | AVG | CAT |
|---|---|---|---|---|---|
| clean accuracy | 57.7% | **61.4%** | 55.1% | 57.0% | 54.0% |
| pixel attack ($\epsilon = 0.03$) | **26.1%** | 5.0% | 23.6% | 22.0% | 23.7% |
| flow attack ($\epsilon = 0.35$) | 23.2% | **34.8%** | 29.1% | 29.9% | 32.0% |
| Union | 19.5% | 5.0% | 22.3% | 20.9% | **22.9%** |
| Composite (0.5) | 24.1% | 3.4% | 25.2% | 25.1% | **27.2%** |

Table 11. Performance of CAT and baselines on CIFAR100 with respect to pixel and spatial perturbations. Rows represent attacks, and columns denote robust trained models. $P_{\text{flow}}$ and $P_{\text{pixel}}$ are models trained with PGD attacks (with corresponding perturbation spaces).

| Method | Time per epoch (minute) |
|---|---|
| MAX | 63.2 |
| AVG | 63.5 |
| MSD | 33.3 |
| CAT | 24.7 |

Table 12. Computational Cost of CAT and baseline methods.

