# OpenReview forum: "Composite Adversarial Training for Multiple Adversarial Perturbations and Beyond"
_ICLR.cc/2021/Conference — Reject_

### Official Review · AnonReviewer3 · 2020-10-28
**Official Blind Review #3**

**Rating:** 5
**Confidence:** 3

**Review:**

Summary:

This paper proposed an interesting new form of adverserial attack (composite adversarial attack) as well as an algorithm to defend this form of attack (CAT). The new form of attack is constructed as a composition of different individual perturbation models including pixel perturbation and spatial transformations. The CAT is proposed to defend both individual attack and composite attack by penalising the maximum accuracy loss during the sequential generation process of a composite attack perturbation. Empirical experiments comparing the proposed algorithm under both individual and composite attacks are conducted on benchmark datasets against baseline methods. The proposed CAT outperformed the baselines under composite attacks. Further analysis and discussion on different variations of composite attack as well as CAT are also presented with possible future exploration directions.

Pros:
- The paper is well structured in general and easy to understand.
- The idea of composite attack is interesting and meaningful to the neural network adversarial attacking area.
- The proposed method improves the network robustness under composite attack.
- The detailed analysis on composite attacks is valuable.

Cons:
My main concern with the paper is the general performance of the proposed algorithm and the fairness of the comparison.
- While the paper claims outstanding performance on individual perturbation model attacks, it is not always true across the two dataset. And the proposed algorithm always presents a lower clean accuracy in most of the experiment settings by a relatively large margin. There seems to be a clear tradeoff between the clean accuracy and the robustness towards a more aggressive attack (composite attack). The result limits the strength of the algorithm.
- I am concerned about the fairness of the comparison against baseline methods like MAX/ AVG. Since the paper used pretrained models from previous work,  MAX/AVG baseline models are trained based on Eq(2)(3) and evaluated under the composite attack. In this case, the underlying perturbation space considered in Eq(2)(3) is different from (smaller than) the one in composite attack. (E.g. true maximum perturbation will not never be considered when training these models)
- Another question is:  what does alpha mean for baseline methods during training? Is alpha used to rescale the perturbation during baseline training or not? If not, then Figure 4 presents very limited information since alpha is an unfair information available to the proposed model. If yes, then isn’t the whole experiment a scaling version of the main results?


Other comments:
- I would move the introduction of spatial transformation perturbation to section 2 as it is part of the fundamentals.
- Some details of baselines in Appendix A should be moved to the main text to provide a more self-contained experiment section. E.g. how the baseline models are trained.
- It would be nice to bold the best performance number in the tables.


---------------------------------------
post-rebuttal

I would like to thank the authors for their efforts to improve the methods and the draft. Part of my concerns was resolved.
For clean accuracy, CAT-r did provide a better trade-off. However, it is improved after the submission deadline, it can't be counted into the original contribution in theory.
For the concern that the comparison to the baseline presents unfairness as the proposed method was designed for the composite attack with a larger perturbation space, I think the author agrees with my point to some extend.
I decided to keep my original score deal to the remaining weakness in the paper.

---

> ### Author Response · Authors · 2020-11-25
> **Response to AnonReviewer3**
>
> We appreciate comments and concerns from the reviewer.
>
> We've made a few modifications to the draft, with additional results and formatting fix-ups. Below we respond to the concerns and comments from the reviewer.
>
> C1 - performance & clean accuracy: First, we acknowledge that the CAT does not always perform better than PGD, AVG, MAX, and MSD, especially for individual perturbation types. We will fix our tone about individual perturbations systematic in the next iteration or camera-ready version. Second, as for clean accuracy,  we proposed a lightweight trick called CAT-r (r for replacement) to make the final model a better trade-off between clean accuracy and robust accuracy. It improves results on three of all four settings in the paper.
>
> C2 - fairness of the comparison: This is a good point from the reviewer. We argue that CAT works differently from AVG, MAX, and MSD by using the composite adversarial attack. They might have different trade-offs in terms of clean accuracy and the "union" and the "composite" accuracy. Plus, the additional results in C1 indicate that CAT-r can achieve better clean accuracy and better robust accuracy than naive CAT.
>
> C3 - $\alpha$ for baselines: No. $\alpha$ is not used in training baseline methods. The point here is to measure each model's robust accuracy to varying $\alpha$. We will make further edits to make the results more accessible.
>
> OC3: As suggested by the reviewer, the results are bold now.

---

### Official Review · AnonReviewer2 · 2020-10-28
**Nice contribution to adversarial training literature but with mixed results**

**Rating:** 5
**Confidence:** 3

**Review:**

The authors propose a method for dealing with *composite* adversarial attacks, which are defined as a sequence of perturbation operators each applying some constrained perturbation to the output of the previous operator. Their method models the composed adversarial examples $x^*$ as the sum of the unperturbed example with a series of perturbations $\delta_i$ which maximize the estimator's loss. They compare their results to other existing adversarial training methods against multiple types of adversarial attacks.

Pros:
- Interesting idea, seems like a very natural continuation of existing work
- Good experimental design, results are reasonably thorough
- Some results are encouraging

Cons:
- Explanation of method (CAT) is somewhat lacking. It's not clear to me exactly what their method does differently than the baselines explained in the background.
- Results are mixed with discussion focusing almost entirely on the positive parts. For example, CAT consistently performs significantly worse than baselines on "clean accuracy" and worse than one or more baselines on other singular attacks (see Tables 1,2,3,4).
- Results in section 5.2 lack explanation (i.e. what do the table columns/rows actually mean)
- Minor formatting issues

Overall, I think the central problem that the authors are trying to solve is important and their work makes a reasonable contribution towards the solution. Despite the apparent mixed results, this paper should be a candidate for acceptance.

Additional comments for the authors:
- It would be helpful to provide references for the definitions of "robust accuracy" and "clean accuracy"; I'm sure these are metrics that have been defined and used in prior work but this can sometimes make it difficult for outside readers to find where they are rigorously defined.
- As mentioned in the Cons, you should make it more clear what the reader should be looking for in the tables. Reading just by the accuracy scores, it seems like CAT often performs worse or about the same as baselines in multiple experiments.
- Table captions should be above, not below, the table. This particularly problematic with Table 4/Figure 4 where the Table caption looks like the title of Figure 4.
- As mentioned before, equation 8 does not (for me) satisfactorily explain what CAT actually does.
- In equation 8, $\delta_i$ appears in the constraint but not in the expression; perhaps you meant to write:
$$
x' = \underset{x^{(m)}}{\arg\max} \ell (f_{\theta}(x^{(m)} + \delta_i,y)
$$
- The distinction between the different indexing notations $x_i$ and $x^{(i)}$ is not always clear
- It's not clear what the notation means in Tables 5, 6, and 7 and how it relates to "ordering" of perturbations.

---

### Official Review · AnonReviewer4 · 2020-10-28
**Official Blind Review #4**

**Rating:** 6
**Confidence:** 3

**Review:**

Summary

This paper proposes adversarial training with a novel threat model. Specifically, the authors propose to compose multiple adversaries, such as the ones based on l_p norm and spatial transform, in a predefined order to create a strong adversary in the adversarial training. The paper empirically demonstrated that the composite adversary is effective against previous adversarial defense mechanisms. It also demonstrated that the proposed adversarial training can lead to the classifier robust against the composite attack as well as the individual or union of multiple adversaries.

Pros
+ The composite adversary seems to be novel and effective in terms of both adversarial attack and adversarial training.
+ The paper is generally well-written and easy to read.
+ The experiment results convey comprehensive evaluation and analysis. I especially enjoyed that it covers various attack scenarios, such as the ones with unseen attacks and composite attack with a random order, etc.

Concerns & Suggestions
- It is not clear why the composited thread model can be stronger than individual or union attacks as claimed by the authors. If there are some theoretical justifications/proofs, it would be interesting to see such discussions (e.g., the composited attack consistently leads to higher classification loss (inner maximization of adversarial training objective)).
- Although I appreciate authors for their comprehensive experiments, the current results are based on fairly small and easy datasets and it would be still interesting to see the results on more complex datasets such as Cifar-100 or mini-ImageNet.
- It is unclear how exactly the l_p attacks are implemented. In Section 4.1, the authors mentioned various methods for l_p attacks, such as PGD, FGSN, C&W, DeepFool, Salt&Pepper, etc., but it is unclear how they are actually used in the experiments, for instance in Table 1 and 2.
- It would be clear if authors add constraints on total attack budget on Eq.(8)
- Table 5 & 6: Please clarify that the rows are the adversarially-trained models and columns are threats. It is confusing since rows and columns are different from the previous tables.

--- post rebuttal update ----

The authors successfully addressed my initial concerns regarding more analysis and experiments on a larger dataset. Therefore, I keep my rating weak accept.

---

> ### Author Response · Authors · 2020-11-24
> **Response to AnonReviewer4**
>
> We thank the reviewer for the encouraging feedback.
>
> We've made modifications and additional experiments to improve our draft. Here we provide our responses to each concern and suggestion.
>
> P1 - Threat model: We add a short theoretic analysis in Appendix A.
>
> P2 - Larger datasets: We've added preliminary results for the CIFAR-100 dataset in Appendix D.3. The results show a similar trend as in the CIFAR-10 case. In the original version, we only performed experiments on MNIST and CIFAR10 since the baselines (MAV, AVG, and MSD) just consider these two datasets.
>
> P3 - Implementation of $\ell_p$ attacks: We follow the default hyperparameters used in the Foolbox for these $\ell_p$ attacks. We've added a description in the "result" part of Section 4.1 to tell readers how results of individual attacks are aggregated into each row in Table 1, 2.
>
> P4 - Equation 8: We rework it as Equation 7 in the revised version.
>
> P5 - Presentation of results: Following the reviewer's suggestion, we replenish the detail in the caption of Table 1, 2, 3, 4.

---

### Official Review · AnonReviewer1 · 2020-10-30
**[Official Review]**

**Rating:** 5
**Confidence:** 4

**Review:**

#### Summary ####
This paper tackles the problem of adversarial training for the image classification task. It proposed a novel adversarial training method called composite adversarial training (CAT) against combined attacks constructed by multiple perturbations. First, CAT is based on the composite adversarial attacks, in which the attackers explore different sources of perturbations. Second, CAT leverages the composite adversarial attacks as the inner loop for optimization during the training. The experimental evaluations have been focused on comparing the proposed CAT with existing robust training methods including adversarial training with PGD attacks, AVG, MAX (Tramer and Boneh, 2019), and MSD (Maini et al. 2020) on MNIST and CIFAR-10 classification benchmarks.

#### Comments ####
This paper studies an important problem in adversarial machine learning. The paper is well-motivated with novel technical contributions (Section 3.1) supported by reasonably designed experiments. However, reviewer feels the submission in the current form is a borderline case mainly due to mixed or inconclusive experimental results.

W1: The clean accuracy of CAT  (Table 1 - 4, first row, last column) is significantly worse than methods such as AVG & MAX and MSD, especially on CIFAR-10 where the accuracy drops 20+% (I assume the state-of-the-art model has 90+% accuracy for the 10-way classification on CIFAR-10). This seems to be a major weakness of the proposed method. Reviewer understands the tradeoff between clean accuracy and accuracy under attack, but not sure how much value it is given the proposed defense method sacrifices too much on the clean accuracy. What makes it worse, this is just the performance drop of 10-way classification on CIFAR dataset. Reviewer is worried if this gap is even more significant on CIFAR-100 or ImageNet (w/ 1000 classes). It would be good to have some ablation studies.

W2: Besides the drop on clean accuracy, reviewer fails to see a clear winner between MSD and CAT (see the last two columns in Table 1 and Table 2). CAT seems to be more robust to composite attacks but not as robust as MSD on other attacks. Such comparisons are missing in Section 4.2 (pixel perturbation and spatial transformations). It would be good to comment on this.

W3: It would be good to report the computational cost (e.g., number of iterations in optimization, training time) of the proposed composite training method and explain how it is compared to the existing methods.

Minor1 (applied for all the tables): it would be good to mention each row is a different attack method and each column is a different defense (robust training) method. It is not crystal clear at the first glance.

---

> ### Author Response · Authors · 2020-11-24
> **Response to AnonReviewer1**
>
> We first thank you for the detailed comments from the reviewer.
>
> We've made modifications and additional experiments to improve our draft. Here we provide responses that try to answer and address each comment.
>
> W1: Regarding drops in clean accuracy, we proposed a lightweight trick called CAT-r (r for replacement) to make the final model a better trade-off between clean accuracy and robust accuracy. It improves results on three of all four settings in the paper. Besides, in Appendix D.3, we supply partial results for the CIFAR-100 dataset. Table 11 and Table 12 show that CAT, MAX, AVG has a similar impact on the clean accuracy even on a complicated dataset. The worse clean accuracy of CAT, MAX, AVG compared to $\ell_p$ PGD models may cause by the inherent challenges in achieving robustness for diverse adversaries.
>
> W2: CAT indeed fails to outperform MSD in some cases. However, we argue that the point of CAT is to achieve robustness in both the previously defined union threat model as well as the new composite threat model. The missing of MSD in Section 4.2 is that MSD requires the perturbations of all threats to be expressed in the same space. While this is straightforward in $\ell_p$ cases, it is hard to unify spatial and pixel perturbation together in the same space.
>
> W3: We add Table 12 in Appendix D.4 to show CAT's computational cost and baseline methods. The comparison is made under the same batch size and the same total number of epochs. Therefore, the result indicates that the CAT is fast.
>
> Minor1: we introduce how rows and columns in Table 1, 2, 3, 4 work in the caption of each Table.
>
> Though it is a bit late, we're looking forward to hearing more comments and suggestions from the reviewer.

---

### Author Response · Authors · 2020-11-23
**Summary of Revision**

We are grateful for the encouraging and insightful comments from reviewers. We've made a few modifications to the draft. Here we summarize the changes we made:

1) Efforts on improving clean accuracy
   In Section 3.2. We add CAT-r, a lightweight trick to fulfill a better trade-off between clean accuracy and robust accuracy against the union and composite adversary. Its results are appended to the last column of Table 1, 2, 3, 4. Now the clean accuracies are closer to MAX/AVG and MSD. Furthermore, the performances are still better MAX/MSD/AVG for Table 2, 3, 4. It is quite close to the MSD in Table 1.

2) A new CIFAR-100 experiments
  In Appendix D.3, we provide additional results on the CIFAR-100 dataset. The results show a similar trend as Table 1, 2, 3, 4. There is a large accuracy gap between MSD/AVG/CAT to clean model. The gap between CAT to MAX/AVG is less than 10%, as in other tables, which indicates that our CAT does not scale worse concerning dataset size. Therefore, we believe there might be an inherent trade-off between adversarial robustness for multiple perturbation and clean accuracy.

3) Computation efficiency comparison
  In Appendix D.4, we compare the computational efficiency of CAT to MAX, AVG, and MAX. The results show CAT is much faster and is able to work with a larger dataset.

4) Justifications for composite attack
  In Appendix A, we analyze the strength of union and composite attacks under a simple setting.

5) Formatting & Descriptions
   - Marks attack and robust models in the caption of Table 1, 2, 3, 4.
   - In Section 4.1, we add how each p norm's attack results are aggregated and what individual perturbations contribute to the result.
   - We bold the best results in Table 1, 2, 3, 4.
   - Equation 8 in original version is reworked as Equation 7 in the revised version.

We'll post our responses to each reviewer right away.

---

### Decision · Program_Chairs · 2021-01-07
**Final Decision**

**Decision:**

Reject

**Comment:**

I thank authors and reviewers for discussions. Reviewers found the paper (specially the CAT-r method proposed in the rebuttal period) interesting but there are some remaining concerns about the significance of the results and experiments. Given all, I think the paper still needs a bit of more work before being accepted. I encourage authors to address comments raised by the reviewers to improve their paper.

- AC